# Combined NMR and molecular dynamics conformational filter identifies unambiguously dynamic ensembles of Dengue protease NS2B/NS3pro

Tatiana Agback [1,7], Dmitry Lesovoy[2,3,7], Xiao Han[4], Alexander Lomzov [5], Renhua Sun [4], Tatyana Sandalova [4], Vladislav Yu. Orekhov[3,6], Adnane Achour [4✉] & Peter Agback [1✉]

The dengue protease NS2B/NS3pro has been reported to adopt either an 'open' or a 'closed' conformation. We have developed a conformational filter that combines NMR with MD simulations to identify conformational ensembles that dominate in solution. Experimental values derived from relaxation parameters for the backbone and methyl side chains were compared with the corresponding back-calculated relaxation parameters of different conformational ensembles obtained from free MD simulations. Our results demonstrate a high prevalence for the 'closed' conformational ensemble while the 'open' conformation is absent, indicating that the latter conformation is most probably due to crystal contacts. Conversely, conformational ensembles in which the positioning of the co-factor NS2B results in a 'partially' open conformation, previously described in both MD simulations and X-ray studies, were identified by our conformational filter. Altogether, we believe that our approach allows for unambiguous identification of true conformational ensembles, an essential step for reliable drug discovery.

[1] Department of Molecular Sciences, Swedish University of Agricultural Sciences, PO Box 7015, SE-750 07 Uppsala, Sweden. [2] Department of Structural Biology, Shemyakin-Ovchinnikov, Institute of Bioorganic Chemistry RAS, 117997 Moscow, Russia. [3] Swedish NMR Centre, University of Gothenburg, Box 465, 40530 Gothenburg, Sweden. [4] Science for Life Laboratory, Department of Medicine, Karolinska Institute, and Division of Infectious Diseases, Karolinska University Hospital, SE-171 76 Stockholm, Sweden. [5] Laboratory of Structural Biology, Institute of Chemical Biology and Fundamental Medicine SB RAS, 630090 Novosibirsk, Russia. [6] Department of Chemistry and Molecular Biology, University of Gothenburg, Box 465, 40530 Gothenburg, Sweden. [7]These authors contributed equally: Tatiana Agback, Dmitry Lesovoy. ✉email: adnane.achour@ki.se; Peter.agback@slu.se

nfection by any of the four Dengue virus serotypes DENV1-4 can lead to Dengue fever, as well as to the significantly more severe Dengue haemorrhagic fever and/or Dengue shock syndrome. Appropriate maturation of DENV particles requires the multifunctional protein NS3 which comprises a serine protease domain (NS3pro) that is essential for polyprotein maturation and viral replication, making it an attractive drug target[1–4]. Furthermore, in vitro studies revealed that the adequate function of the NS3pro domain requires the NS2B cofactor segment derived from the viral protein NS2 for full NS2B/NS3pro protease activity[5].

The molecular bases underlying the function(s) of DENV-associated NS2B/NS3pro proteases, and a thorough understanding of the link(s) between overall conformation and function of this heterodimer remain unclear. A thorough understanding of unique or multiple conformations may be an essential step for the adequate development of inhibitory compounds[1,6–8]. Several crystal structures of flaviviral serine proteases have been hitherto determined[3,9], including NS2B/NS3pro from all four DENV serotypes[10–14]. A comparison of these structures revealed that NS3pro always adopts a conserved fold comprising two β-barrels. Furthermore, the conformations of the catalytic residues His51 and Ser135, both localized within the active site, are similar in all four DENV and in several other Flaviviridae proteases, indicating that these structural features are conserved despite differences in sequences. Still, despite these structural similarities, it is well established that the most prevalent DENV-2 is significantly more infectious compared to the three other DENV serotypes[15], indicating that differences in conformations and/or minor localized structural differences may have profound effects on the function of NS2B/NS3pro[16].

In contrast to NS3pro, the overall conformation of NS2B varies between different crystal structures of this heterodimer. Therefore, the hypothetical existence of two distinct conformations for the C-terminus region of NS2B has been previously suggested[17], where NS2B may adopt either a 'closed' conformation as found in a majority of inhibitor-bound complexes, or an 'open' conformation described in crystal structures of NS2B/NS3pro apo-forms[11,17]. In the closed form, NS2B circumflexes more than 300º around the equatorial region of NS3pro, with the C-terminal β-hairpin of NS2B wrapped around the active site of NS3pro (Supplementary Fig. S1a). 'Open' NS2B conformations described in ligand-free flaviviral proteases can be divided in (i) disordered open and (ii) alternative open conformations. A stretch of only about 20 NS2B amino acids is visible in the electron density of proteases with 'disordered open' NS2B conformation (Supplementary Fig. S1b). In the 'alternative open' conformation the positioning of the first 16 NS2B residues is very similar compared to the 'closed' form, while the rest of NS2B folds away from NS3pro, retracting from the active site. Furthermore, the C-terminal stretch of residues 62-96 in NS2B forms a short α-helix followed by a β-strand composed of residues 70/73, resulting in a conformation that has been described as the 'fingerprint' for the inactive 'open' form of NS2B/NS3pro proteases[11] (Supplementary Fig. S1c). It should be noted that $NS2B_{62-96}$ is in close contact with symmetry-related molecules in all crystal structures of NS2B/NS3pro apo-forms, which may explain this alternative 'open' conformation[11,17]. This can either result in high mobility and disorder of sections of NS2B (Supplementary Fig. S1d), or induce a possible artefactual conformation through tight contacts with surrounding molecules in the crystal (Supplementary Fig. S1e). We therefore argue here that the observed 'alternative open' conformation of the C-terminal part of NS2B is most probably induced by crystal contacts rather than describing adequate interactions with NS3pro.

The prevalence of the different possible conformations in solution remains thus elusive. Despite X-ray studies of the apo

form DENV-2 NS2B/NS3pro indicating that the C-terminus of NS2B was disordered[11], other NMR[18,19] and paramagnetic labelling studies[20,21] demonstrated that NS2B adopts predominantly a 'closed' conformation in solution. We hypothesized here that a higher localized flexibility corresponding to the 'open' conformation could be combined with the possibility for intermediate states. If valid, the 'open/closed' hypothesis would require conformational changes to occur upon e.g. activation by substrate or binding to an inhibitor, a concept initially supported by the crystal structure of NS2B/NS3pro in complex with an allosteric inhibitor (PDB code 4M9T)[22], by molecular dynamics (MD) simulations of NS2B/NS3pro in 'open' conformation whence binding an allosteric inhibitor[23–28] and through mutational studies forcing NS2B to assume an 'open' inactive-state conformation[29]. This hypothesis would also imply the simultaneous presence of 'open' and 'closed' conformations, with a possible exchange time scale of milliseconds measured in ref. [30]. Finally, NS2B may also adopts an 'intermediate' conformation as described in the crystal structure of DENV-4 NS2B/NS3pro (PDB code 7VMV)[31].

The combination of NMR spin relaxation spectroscopy with molecular dynamics (MD) simulations represents in our opinion methods of choice to assess the dynamics of biomolecules. Although MD simulations provided information on the motions of all atoms in NS2B/NS3pro in complex with inhibitors[32–34], it should be noted that the results depend on the applied force field and the computational protocols[35]. It is well established that the validation of such MD simulations by experimental results is critically important[36–41]. Based on previous results[40,42], we have developed a protocol for conformation filtering in which MD simulation results were compared with NMR relaxation data. Since the three-dimensional structure of the apo form of DENV-2 NS2B/NS3pro complex in solution remained missing, we performed a detailed NMR investigation starting with the apo form of the Ser135Ala (NS2B/NS3proS135A) mutated protein variant. This mutation abolishes protease activity with minimal interference on the overall three-dimensional structure[43]. We have previously reported a near complete description (>95%) of backbone $^1HN$, $^{15}N$, $^{13}C^{\alpha}$, $^{13}CO$, $^1H^{\alpha}$ and sidechain $^{13}C^{\beta}$ chemical shift assignments for DENV-2 NS2B/NS3proS135A[44]. We also assigned methyl resonances for the side chains of valine, leucine, and isoleucine residues.

Our results demonstrate that the choice of adequate ensembles of potential conformations for large proteins, combined with the use of force fields suitable for the task and stringent NMR data allows us to unambiguously probe the existence of different NS2B/NS3pro conformations. We assessed the binding pocket flexibility of NS3pro and the intrinsic flexibility of NS2B, and our results show the existence of mainly closed conformations, with relatively small and localized conformational shifts in limited sections of NS2B. Our results, based only on unlinked full-length NS2B/NS3pro protease heterodimers, demonstrate that the open inactive form of NS2B/NS3pro is not present in the solution, and that this alternative conformations rather appears to be mainly due to crystal packing. We believe that our approach can provide the scientific community with a more reliable template for the future development of inhibitory compounds, and that our protocol can be used for the unambiguous identification of different conformations for large proteins in solution.

## Results

**Development of a protocol for NMR-restrained MD simulations.** To optimally explore the conformational space and generate an ensemble of conformations, series of 100 simulated annealing (SA) steps with NOE-based restraints were performed

using as a starting point the three-dimensional structure of the 'closed conformation' of the DENV-2 NS2B/NS3proS135A heterodimer, which was created by homology modelling using SwissModel (https://swissmodel.expasy.org/)[45], and the crystal structure of DENV4 NS2B/NS3pro (PDB code 5YVU) as a template. Missing residues were added including the NS3pro N-terminal His-tag, and differing DENV-4 residues were replaced to the corresponding amino acids in DENV-2 (Fig. 1a) using UCSF Chimera 1.15[46]. The MD simulation protocol for initial system preparation and equilibration is described in Supplementary Note S1. Our SA protocols included heating for 0.4 ns

from 300 to 500 K, followed by a cooling step to 300 K for 0.1 ns. Altogether, 366 NOE distances and 334 torsion angle restraints were applied with force constants 20 kcal/mol/Å² and 2 kcal/mol/rad², respectively (Supplementary Table S1). After SA analysis, three ensembles of NS2B/NS3pro conformations were selected, including one structure with a well-formed β-hairpin in NS2B, and two structures in which the positioning of the NS3 N-terminal his-tag differed (Supplementary Fig. S2). They were all subjected to 1μs MD simulations with similar NMR NOE distance and angle restraints. The analysis of the ten molecular models obtained by cluster analysis following the NMR-

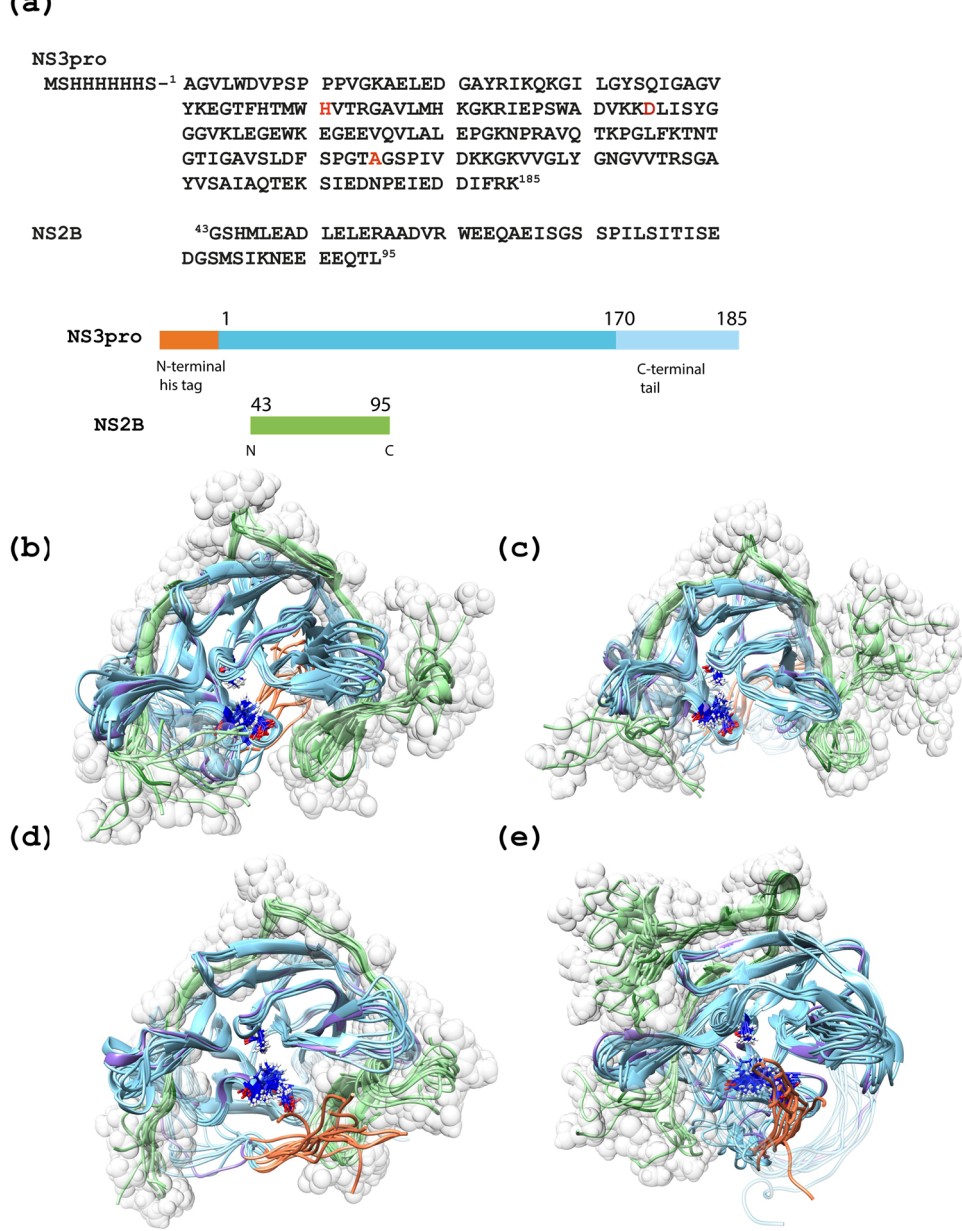

**Fig. 1 The four main conformation ensembles obtained using free restraints MD simulations for DENV2 NS2B/NS3proS135A. (a)** The sequences of the NS3proS135A and NS2B domains are displayed with corresponding numbering and schematic colours used in the structural models presented in (**b**)–(**e**). The catalytic triad comprising His51, Asp75 and the mutated S135A residues are indicated in dark blue. **b**–**e** Ribbon representations of the domain structures of four ensembles comprising 10 obtained molecular models of the DENV-2 NS2B/NS3proS135A heterodimer are presented, including: (**b**)–(**e**) ensembles I, II, III and IV, respectively. The globular NS3proS135A domain, comprising residues 1–170, is coloured in blue. N-terminal his tag is coloured in coral. The disordered NS3proS135A C-termini tail is coloured in light blue. The starting structures used for free MD simulations presented in (**b**)–(**e**) were obtained as described in the Material and Method section, and are displayed in indigo and dark green for the NS3proS135A and NS2B domains, respectively. The surfaces corresponding to the Van der Waals radius of each heteroatom in the co-factor NS2B are displayed transparent white-grey.

restrained 1μs MD trajectory of the three conformation ensembles, I*, II* and III* are presented in Supplementary Note S1, Supplementary Table S2 and S3 and Supplementary Fig. S2. Importantly, based on the observed NOE in the NOESY spectrum of DENV-2 NS2B/NS3proS135A, seven of the nineteen identified intermolecular distances between NS3pro and NS2B in the range 109-117 and 68- 88 aa, respectively, were below 5 Å. These distances are in agreement with the 'closed' but not the 'open' conformation. Indeed, the corresponding distances in the 'open' conformation calculated from the crystal structure of DENV-2 NS2B/NS3pro (PDB code 2FOM) exceed 7.7 Å. Applying SA protocol with NOE-based restraints converted the 'open' conformation to a partially 'closed' conformation ensemble. Instead it was used here directly as reported in 2FOM structure to the free MD simulation described below[47].

**Identification of separate structural ensembles based on free MD calculations.** Three structures obtained at the end of 1μs MD trajectories with NMR restraints from each of the three identified conformation ensembles I*, II* and III* were thereafter selected as starting structures for free 1 μs MD simulations, creating novel conformations ensembles I, II and III. Ensemble I and III have a well-structured β-hairpin formed by residues 76–86 whereas for ensemble II this region is more flexible. In ensemble III the N-terminal his-tag at one end of the NS3pro domain points towards the C-terminal part of the NS2B co-factor.

Although not identified through the protocol described above and in the supplementary material section, a fourth MD trajectory (called from now on IV) of an 'open' conformation was also obtained by starting from the crystal structure of the 'open' DENV-2 NS2B/NS3proS135A (PDB code 2FOM)[11]. RMSD of backbone heavy atoms for dynamically stable residues 20–168 of NS3proS135A and 51–72 NS2B reached the plateau after up to 50 ns of free simulation for all ensembles, whereas all residues became stable after 500 ns (Supplementary Fig. S3). Thus, the last 500 ns were taken for further analysis. Altogether, 10 final structures were obtained by cluster analysis from free MD simulation trajectories for each of the four conformational ensembles I–IV (Fig. 1). Thereafter these MD simulated trajectories were used for back-calculation of the relaxation parameters for NS2B/NS3proS135A.

**Review of relaxation parameters used for side chain methyl evaluation dynamics.** To select the appropriate relaxation parameters to be evaluated and experimentally measured is crucial to this study, due to the time consuming computation processes, combined with extensive NMR experimental time. Side-chain methyl dynamics in MD simulations were verified using conventional $R_1$ and NOE, as well as dipole/dipolar (denoted as Γ1[48] and Γ2) and CSA/dipolar cross-correlation contributions to $R_1$ and $R_2$ (denoted by H1 and H2), respectively. We initially performed simulations of all the six relaxation parameter sets for methyl groups including $R_1$, NOE, Γ1, Γ2, H1 and H2 (Supplementary Figs. S4, S5), using free MD trajectories for each different conformational ensemble (Fig. 1). Additionally, using the 'closed' conformation as a starting structure, identical to conformational ensemble I, the MD simulations were repeated with NMR restraints (called structural ensemble V). This additional step was performed to assess the influence of additional restraints in MD simulations on the obtained theoretical relaxation parameters as a test of force field perturbation. A value of $2.08e^{-08}$ s was used for the overall correlation time $\tau_c$ in all calculations, based on our isotropic tumbling model analysis (Supplementary Note S2). Importantly, the transverse relaxation rate $R_2$ for methyl groups was not included because these could not be reliably measured by

NMR due to dipole/dipolar cross-correlation effects[42,49,50]. Careful perusing of our results presented in (Supplementary Figs. S4, S5), of the $R_1$, NOE, Γ1, Γ2, H1 and H2 relaxation parameters for the different trajectories from the free MD calculations allowed us to conclude that $R_1$ and Γ2 are the most sensitive values to evaluate MD trajectories profiles. These two parameters represent therefore the best sensors for conformational balance as well as potential differences among MD trajectories (Supplementary Figs. S4, S5). Additional criteria that were used within the frame of this study included the simulated curves (Supplementary Fig. S4), where the relaxation parameters $R_1$, NOE, Γ1, Γ2, H1 and H2 were correlated with the internal motion of the methyl group $\tau_e$ and its amplitude S2 (Supplementary Fig. S6). According to the results from these simulations, the $R_1$ and Γ2 relaxation parameters complement each other very well, showing higher sensitivities to $\tau_e$ and S2, respectively. In contrast, all the other relaxation parameters from methyl groups displayed less stringent features. Indeed, the NOE parameter was not sensitive to conformation and sequence diversities for both NS3proS135A and NS2B, and could fit to multiple values of $\tau_e$ and S2 (Supplementary Fig. S4b). Variations for H1, H2 and Γ1 are limited in narrow ranges 0.0 to $-0.5 \, s^{-1}$, 0.0 to $-1.2 \, s^{-1}$ and 0.0 to $-1.5 \, s^{-1}$, respectively. Moreover, H1 and H2 back-calculated values strongly depend on the CSA parameter for every amino acid residue, which is not always readily available. Hence the $R_1$ and Γ2 relaxation parameters were chosen for further comparison with the obtained corresponding experimental data.

**Validation of structural ensembles based on backbone relaxation data.** Using the four simulated trajectories I–IV with identical lengths obtained from free MD simulations (Fig. 1), the back correlation functions and dynamic parameters $R_1$, $R_2$ and NOE of all the $^1H$-$^{15}N$ amide backbone were calculated according to the developed protocols (See material and method section). Back-calculated theoretical versus experimental $^{15}N$ $R_1$, $R_2$ and NOE parameters for the backbone of NS2B/NS3proS135A at 600 MHz are presented for all five trajectories in Fig. 2. Furthermore, the theoretical profiles of relaxation rates simulated from different trajectories as functions of the internal motions $\tau_e$ and amplitude $S^2$ are also presented in Fig. 2.

The first conclusion that can be derived from the results presented in Fig. 2 is that the dynamic parameters $^{15}N$ $R_1$, $R_2$ and NOE (both back-calculated and experimental) for NS3proS135A residues 20–28 and 35–60 (Fig. 2a–c) and for the NS2B residues 50-64 (Fig. 2d–f) are not significantly different for all five trajectories. Thus, there are no systematic shifts between theoretical and experimental data, a conclusion that supports the validity of our back-calculation protocol presented in this study. Additionally, the S2 and $\tau_e$ parameters as qualitatively estimated from the $R_1$, $R_2$ and NOE parameters (Fig. 2), are larger than 0.8 and 150 ps, respectively, which is in agreement with the corresponding parameters obtained with the Lipari and Szabo model[51,52] (Supplementary Figs. S7d–h). Altogether these results clearly indicate a continuity of association between the N-terminal segment of NS2B (until residue Ser72) and the NS3 domain. The flexible nature of the backbone in some regions of NS3proS135A and NS2B is also revealed by the relaxation analysis, as identified by lower values for the heteronuclear NOE and $R_2$ (Fig. 2b, c, e, f), respectively. As expected, reduced NOE and $R_2$ values were observed for the N- and C-terminal residues 1–18 in NS3proS135A, 45-53 in NS2B, as well as for residues 170-185 in NS3proS135A and 86-95 in NS2B. It should also be noted that, clearly lower values for heteronuclear NOE were detected within the loop region linking the β1- and β2–strands spanning

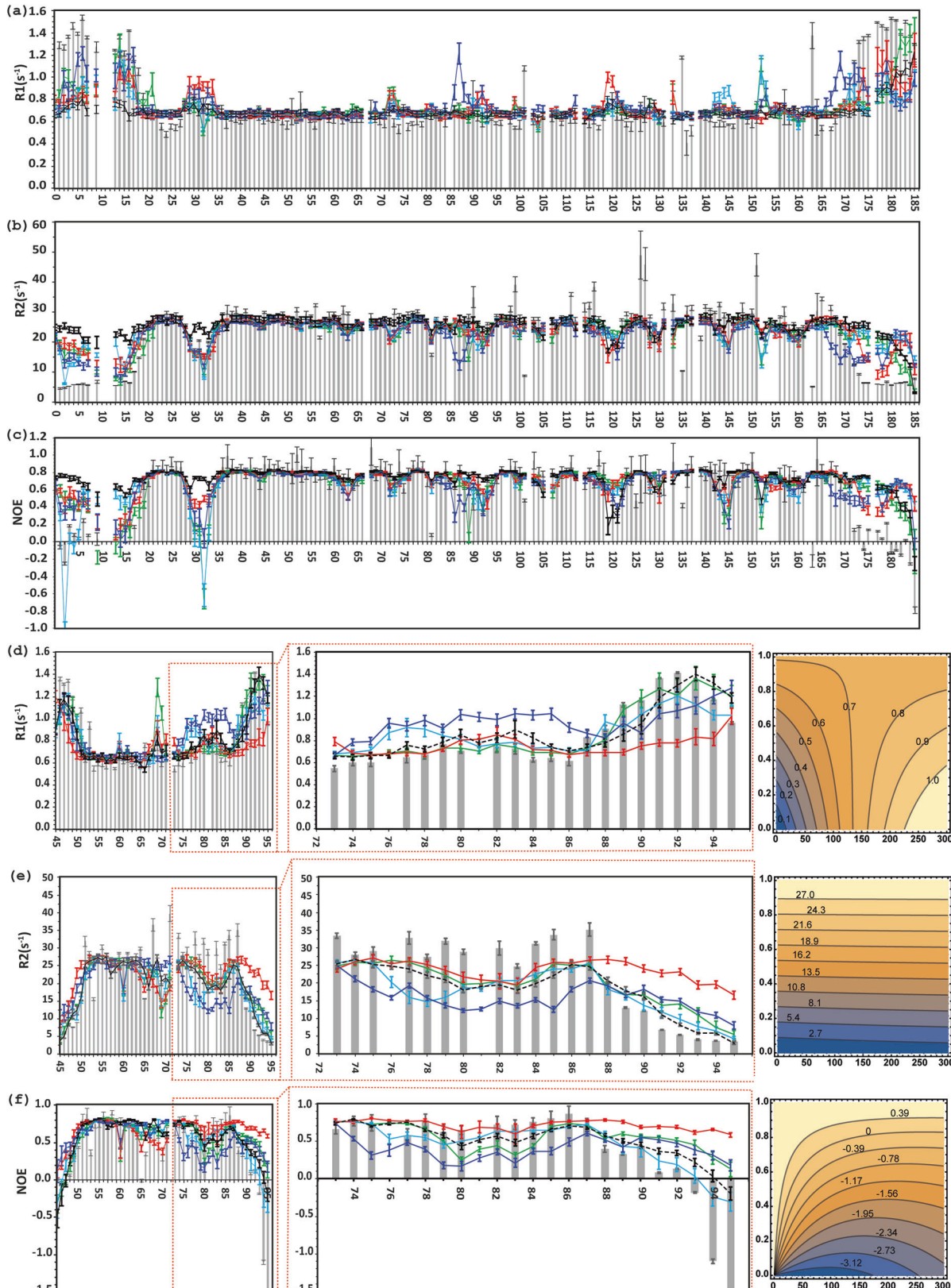

from residues Lys28 to Ser34 in NS3proS135A, indicate its flexibility (Fig. 2c). The fast motions observed within this region were also corroborated by a substantial decrease in $R_2$ values (Fig. 2b) to ca 15 s$^{-1}$ and an increase in $R_1$ values to ca 1.5 s$^{-1}$ (Fig. 2a), indicating that these structural components exhibit extensive fast motion of random-coil on sub-nanosecond timescales. Importantly this observed 'drop' can be reproduced by back-calculation in all four MD trajectories free of restraints.

However, the back-calculated relaxation parameters $R_1$, $R_2$ and NOE for the molecular models corresponding to structural ensembles I-IV display clear differences compared to the fitting of the experimental data for NS2B starting from residue 72 until the

**Fig. 2 DENV-2 NS2B/NS3proS135A amide backbone, $^{15}N(H)$, dynamic parameters of the NS3proS135A and co factor NS2B obtained on 600 MHz spectrometers.** The relaxation parameters for the NS3proS135A (**a**–**c**) and NS2B (**d**–**f**) are presented according to the following: Longitudinal relaxation rate $R_1(s^{-1})$ for NS3proS135A and NS2B are presented in (**a**) and (**d**), respectively. The transverse relaxation time $R_2(s^{-1})$ for NS3proS135A (and NS2B are presented in (**b**) and (**e**). Heteronuclear $^1H–^{15}N$ NOE values for NS3proS135A and NS2B are shown in (**c**) and (**f**). The experimentally obtained $R_1(s^{-1})$, $R_2(s^{-1})$ and NOE values are presented by grey solid brackets. The theoretically predicted dynamic parameters $R_1$, $R_2$ and NOE, obtained through five trajectories, are displayed by solid lines and coloured in green, light blue, red, dark blue and dashed black for ensembles I, II, III, IV and V, respectively. NS2B regions corresponding to residues 72-95 are extended in boxes shown with red point lines within (**d**–**f**). The theoretical profiles of relaxation rates simulated as a function of overall correlation tumbling $\tau_c$, internal motion $\tau_e$ and amplitude $S^2$ are presented in the right panels of NS2B. The error bars of the experimental data are one $\sigma$ from the curve fitting and for the predicted parameters from the bootstrapping analysis.

end of the C-terminal region (Fig. 2d–f). The largest inconsistencies for experimental $R_1$, $R_2$ and NOE were observed for the MD simulation trajectory based on the crystal structure of the 'open' conformation. Indeed, discrepancies between residues 72 and 88 in $R_1$ reach here up to 0.4 $s^{-1}$, and up to 10–15 $s^{-1}$ in $R_2$, with a reduction of the predicted NOE values. These results imply a higher mobility of the C-terminal part of NS2B compared to the overall structure of the heterodimer, and therefore demonstrate the absence of an 'open' conformation in solution, or at best in an extremely low amount. It should also be noted that back-calculated $R_2$ values for all trajectories were ca 5 $s^{-1}$ lower compared to the corresponding experimental values for the stretch of residues 77–87 in NS2B. One potential reason underlying this result could be that back-calculated $R_2$ values reproduce a pure dipole-dipole relaxation term, and that the contribution of the slow exchange Rex term is omitted.

**Validation of structural ensembles based on side chain methyl dynamics.** The methyl longitudinal relaxation rate $R_1$ and the cross-cross correlated relaxation rate $\Gamma_2$, both obtained at 800 MHz, are presented by grey solid brackets in Supplementary Fig. S8. The values for the $R_1$ and $\Gamma_2$ parameters span from ca 4.0 to 0.0 $(s^{-1})$, and from ca 15.0 to 0.1 $(s^{-1})$, respectively, in NS3proS135A (Supplementary Fig. S8a, b), covering a large range of the internal correlation time $\tau_e$ and order parameters $S^2$ (Supplementary Figs. S4a, S4e). Based on these results, the heterogeneity in behaviour of both the $R_1$ and $\Gamma_2$ profiles for NS3proS135A is evident. Nevertheless, all four trajectories that correspond to each of the conformation ensembles I-IV (Fig. 1b–d are in agreement with the experimental data within the range of experimental uncertainties). We consider this result as remarkable in itself but also as unexpected, as it demonstrates that the position/conformation and dynamics of the NS2B co-factor have a minimal influence on the fast dynamics of the NS3proS135A domain.

The situation is different for NS2B where the heterogeneity of the $\Gamma_2$ parameters is notable (Supplementary Fig. S8c, d). Indeed, $\Gamma_2$ values for residues Ile67, Ile73 and Leu(CD1)74 are almost at zero as long as $\Gamma_2$ keeps on ca 10 $s^{-1}$ for all other methyls of NS2B. This indicates that for Ile67, Ile73 and Leu(CD1)74 methyl the $S^2$ order parameters are dropping down according to estimation (Supplementary Fig. S4e). A similar trend was observed for the $R_1$ parameter for NS2B, where $R_1$ values for Ile67 and Ile73 were reduced to ca 1 $s^{-1}$, indicating that the internal correlation time $\tau_e$ values are below $20 \times 10^{-12}s$ (Supplementary Fig. S4a). Furthermore, the back-calculated data for methyl side chains varied clearly in the trajectory of ensemble V ('closed' conformation model obtained with NMR restraints during MD calculations), compared to the corresponding experimental data for both NS3proS135A and NS2B (Supplementary Fig. S8). We consider this result important from a methodological point of view, as it indicates that free MD dynamics reproduces seemingly more correctly the dynamics of methyl side chains for Val, Leu and Ile residues compared to constrained MD calculations.

In order to identify the methyl groups that are mostly affected by interactions formed between NS2B and NS3proS135A, we combined the methyls for Val, Leu and Ile residues into four different hydrophobic clusters denominated Cl(1)-Cl(4) (Fig. 3a, b). The relaxation parameters $R_1$ and $\Gamma_2$ of the methyls in cluster Cl(1) are presented in Fig. 3c, g, respectively. In this cluster differences between experimental and back-calculated $R_1$, $\Gamma_2$ data for the open (IV) but not the closed trajectories (I-III) are observed. Nevertheless, any conclusive choice between the trajectories is ambiguous due to the lack of a few experimental relaxation $R_1$, $\Gamma_2$ data for methyls Val126 and Leu128. The relaxation $R_1$, $\Gamma_2$ of methyls belonging to cluster Cl(2) are presented in Fig. 3d, h. For the four trajectories I-IV, back-calculated values of $R_1$ and $\Gamma_2$ fit well with the corresponding experimental relaxation $R_1$ and $\Gamma_2$ data. This is expected for cluster Cl(2) because it is located in the first β-barrel close to the NS2B and NS3proS135A conservative site of interaction. The relaxation $R_1$, $\Gamma_2$ of methyls belonging to cluster Cl(3) are shown in Fig. 3e, i. It is located between the first and the second β-barrels, close to the active site, and thus this cluster could be very important regarding the interactions of NS2B/NS3proS135A with ligand mimicking substrates. The back-calculated $R_1$, $\Gamma_2$ of methyls of four trajectories I-IV are very similar with the experimental relaxation $R_1$, $\Gamma_2$ data. The most informative results were provided by cluster Cl(4) coloured in green in Fig. 3a, b. The relaxation $R_1$, $\Gamma_2$ of methyls belonging to cluster Cl(4) are shown in Fig. 3f, j respectively. Despite the lack of experimental data for methyl Val162 and no difference in the $R_1$ relaxation parameter, the unambiguous differentiation between the four trajectories is observed for the $\Gamma_2$ relaxation parameters of Val154 in NS3proS135A, and Leu74, Ile76, Ile78 and Ile86 in NS2B. The fully 'open' conformation corresponding to trajectory IV displays the worse fit to the data. Indeed, this cluster is formed in Fig. 3b whereas this cluster is divided in the open conformation shown in Fig. 3a.

## Discussion

Structure elucidation of proteins by NMR has traditionally focused on finding the structure with the best fit to the sets of distance and angle restrains extracted from NMR-related parameters. In general, the best fits for structures solved using NMR were reached for folded, stable proteins, resulting in good correlation with those obtained by X-ray crystallography. In contrast, more dynamic proteins have always been a challenge for structural studies. Recently, a highly dynamic nature for the NS2B/NS3pro fold in flaviviral proteases has been proposed[53], based on the analysis of variations in an ensemble of X-ray structures. Moreover, the recently determined crystal structure of DENV-4 NS2B/NS3pro, in which the NS2B cofactor adopted a clearly 'intermediate' conformation (PDB code 7VMV)[31], opened the possibility for a revision of the commonly accepted simplistic two-step concept in which Dengue type proteases adopt only two conformations, either 'open' or 'closed'. Keeping this in mind, the main goal of this study was to assess whether the inter-domain structures of the highly dynamic apo-form of the DENV-2 NS2B/NS3proS135A complex in

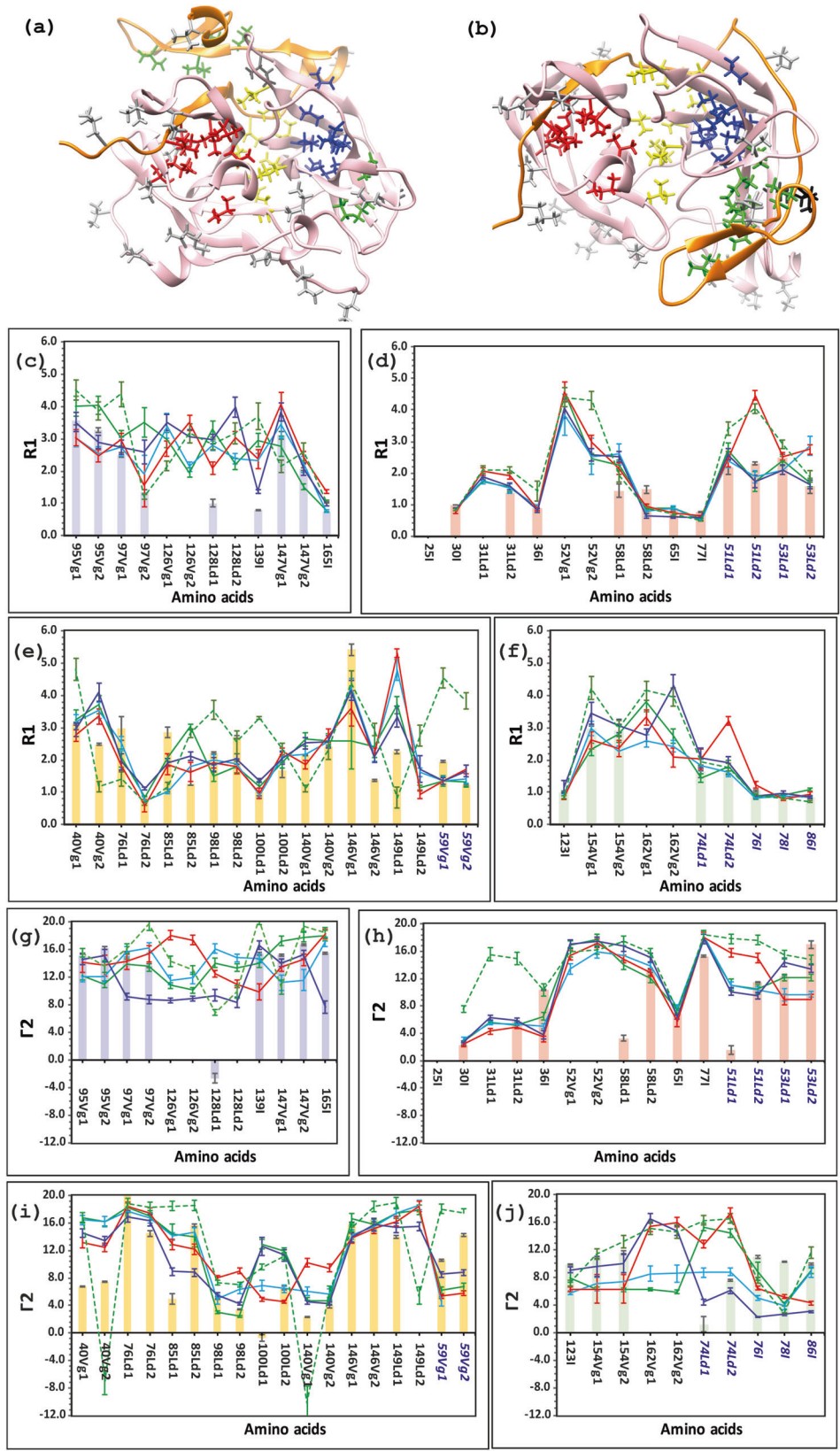

solution as obtained by NMR, differ from the results provided by crystal structure analyses for *e.g.* DENV-4.

To assess the flexibility of NS3proS135A and NS2B, we employed a novel tool by combining NMR spin relaxation spectroscopy and molecular dynamics (MD) computer simulations of dynamic parameters, an approach that was previously developed and successfully applied to dynamic studies of proteins[40,42]. We evaluated if the back-calculated dynamic parameters of different ensembles of the structures obtained in MD simulations, of equal trajectory length but starting from different starting structures, could be ranked according to the best fit to the measured dynamic parameters. Although the choice of starting

**Fig. 3 DENV-2 NS2B/NS3proS135A methyl dynamic parameters of the $R_1$ and $\Gamma2$ obtained on 800 MHz spectrometers.** Experimental values of $R_1(s^{-1})$ (**c–f**) and $\Gamma2(s^{-1})$ (**g–j**) for different methyl clusters in the NS2B/NS3proS135A are presented in blue (**c, g**), red (**d, h**), yellow (**e, i**) and green (**f, j**) solid brackets according to colours on panels (**a, b**). The theoretically predicted dynamical parameters, $R_1$ and $\Gamma2$, obtained through five trajectories, adopted different conformations ensembles shown by solid lines: green (I), light blue (II), red (III), bark blue (IV) and green dashed line (V). (**a**), (**b**) Annotation of the clusters of methyl groups in the DENV-2 mutant S135A. A model of (**a**) 'open' and (**b**) 'closed' forms of the DENV-2 S135A mutant. Four large hydrophobic clusters of methyl Ile, Val, Leu are coloured in blue, red, yellow and green for clusters Cl(1), Cl(2), Cl(3) and Cl(4), respectively. The Methyl of Ile, Val and Leu residues that are located outside of the hydrophobic cores of DENV-2 S135A mutant are coloured in light grey. The error bars of the experimental data are one σ from the curve fitting and for the predicted parameters from the bootstrapping analysis.

structures is arbitrary, it should however cover the conformational space of interest. In this study we created a family of starting structures based on all-atom MD simulations with NMR restraints of the proposed different structures of NS2B/NS3proS135A, i.e. open/semi-open/closed conformations (Fig. 1). The back-calculated relaxation parameters for every ensemble trajectory were compared with those experimentally obtained. Importantly, and advantageously, the choice of relaxation parameters of back bone and side chains of proteins to be measured can be evaluated and restricted. Only those which allow detection of the largest deviations of theoretical back-calculated parameters between different trajectories were chosen. After carefully pursuing the theoretical values for all the ensembles of NS2B/NS3proS135A (Supplementary Figs. S4–S5), the $R_1$, $R_2$, hetero NOE of amide bond and $R_1$, $\Gamma2$ of methyl side chain relaxation parameters, and in particular of the Val, Leu and Ile amino acids, were chosen as sufficient for the task (Figs. 2, 3, S8).

There is still an ongoing discussion about how the applied force field and the used computational protocols can influence the results of back-calculated relaxation parameters[36–41]. We consider this discussion as outside the scope of this publication. Nevertheless, the validity of our back-calculation protocol used in this study was evaluated by comparing the theoretically obtained values against the corresponding experimental relaxation parameters located in the first β-barrel of NS3proS135A, between residues 20–28 and 35–60 (Fig. 2a–c). This approach has recently been reported[54]. All trajectories in this region were expected to show similar results. Indeed, no systematic shifts between theoretical and experimental data were observed. This result also demonstrated the continuity of association between the N-terminal segment of NS2B until Ser72 and NS3proS135A for all investigated ensembles (Fig. 1). Additionally, it is evident that back-calculated parameters follow the heterogeneity values of experimental parameters outside the hot-spot of interactions formed between NS3 and NS2B.

Another issue that has been discussed is that experimentally obtained spin relaxation parameters for both backbone and methyl side chains represent an averaging of values through the superposition of different conformation ensembles. It is in our opinion evident that these parameters are dominated by the ensembles with higher populations. Nevertheless, one can argue that comparing with studies using luminescence assays[55] or $^{19}F$ NMR spectroscopy[56] to monitor conformational transitions, NMR obtained relaxation parameters reflect the local variation for every nuclei of sidechain and backbone[42] directly related to structural changes due to allosteric interactions in proteins or conformational shifts between ensembles. In this study the back-calculated relaxation parameters for three conformation ensembles such as 'closed', 'partially open' or 'partially open with tag' fit to the experimental data (Figs. 2, 3). This means that all three ensembles could be present in an equilibrium. Importantly, the superposition of structurally similar states of the C-termini of NS2B includes the 'partially open' conformation similar to the one observed for DENV-4 NS2B/NS3pro in crystal state. However, the contribution of the ensemble with the 'fully open'

conformation of NS2B observed in X-ray structures should be reduced due to the large violations of the back-calculated *versus* experimental relaxation parameters in the C-termini of NS2B. Indeed, the largest differences are observed for the backbone $^1H$-$^{15}N$ hetero NOE of Glu80 and Ser83 in NS2B stretching ca 0 *versus* 0.6 for the 'fully open' conformation *versus* the experimental values. This result allows us to conclude that if this conformation cannot be fully ruled out as irrelevant only due to the low score of fit, still this possible population can be estimated to be below the level of experimental error, ca 5%. Unfortunately, the present state of MD computation does not allow us to observe the slow transition between higher populated states and the lower populated ones, and this should be considered as a challenge for future development.

The possibility to detect by NMR low populated conformations in the slow exchange limit in proteins is well documented[30,31,57,58]. It should be noted that the 'fully open' conformation is claimed to be a non-active conformation even if it is present according to our results only in a very minor population, if any. This is the opposite of the multiple observations for other proteins in which the 'invisible' minor forms of proteins are 'preferred' as functionally active forms. Keeping this in mind, the results obtained within the present study allow us to propose that the 'fully open' conformation is most likely a crystallisation artefact.

Our approach has also allowed us to establish that the most conserved part of NS2B/NS3proS135A where all four ensembles, including the 'fully open' conformation, fit equally to the experimental relaxation parameters. This is in our opinion a very important and unexpected result due to the reason that NS3proS135A is not properly folded in the absence of NS2B[59]. Nevertheless, the position and interaction in the hot spots between NS2B and NS3 between the four different ensembles do not predict allosteric changes that may be located far away from the contacts. This could be due to that only the apo form of DENV-2 NS3/NS2B was used in the present study. Work is in progress to investigate if the approach presented in this study could be successfully applicable to monitor conformation shifts and changes in the dynamic of backbone and sidechains when the apo form interacts with allosteric ligands or peptide mimic substrates.

Our results indicate that crystal structures of the DENV-2 NS2B/NS3proS135A protease complex may have underestimated its high flexibility, as this protein can adopt multiple conformational states in the apo form in solution. Here we performed a detailed NMR study in solution of the apo form of the Ser135Ala mutated protein variant which abolishes functional activity with minimal interference on the overall three-dimensional structure. We employed a novel, to our best knowledge, newly developed tool, where back-calculated dynamic parameters obtained from free MD simulations of different trajectories of conformation ensembles were compared with experimentally obtained NMR relaxation data of the backbone and side chains of the protein. We expanded on the conformational changes occurring in DENV-2 NS2B/NS3proS135A to three conformational ensembles

obtained in long-timescale MD simulations equally satisfied with the NMR restraints. The three structures representing these ensembles showing differences in the C-termini of NS2B, called 'closed', 'partially opened' and 'partially-open with-tag', and one additional 'fully open' X-ray structure, were submitted to the follow up of free MD simulations of similar length of trajectory. $R_1$, $R_2$ and hetero NOE of the amide bonds and $R_1$ and $\Gamma 2$ relaxation parameters of methyl groups were measured and theoretically predicted. First, we demonstrated the validity of our back-calculation protocol by comparing the theoretically obtained with the corresponding experimental relaxation parameters located in the first β-barrel of NS3proS135A, where no systematic shifts were observed and all trajectories in this region were expected to show similar results. We established an almost perfect correlation between all predicted *versus* experimental dynamical parameters for the three all-atom MD trajectories of conformations known as 'closed', 'partially open' and 'partially open with tag', indicating that they mainly contribute to conformation ensembles of NS2B in solution. Our results reveal that the main discrepancies in the C-termini of NS2B were observed for the trajectory of the 'fully open' conformational ensemble predicted in X-ray studies, indicating that if it exists in solution, it would correspond to a very low population. This allows us to argue that the altered conformation taken by NS2B in the crystal structure of DENV-2 NS2B/NS3proS135A protease complex observed in 'fully open' conformation is most probably due to crystal packing (Fig. S1). We believe that our approach to verify back-calculated relaxation parameters for chosen conformational ensembles provides a possibility for searching more realistic models that can be used for inhibitor screening and modelling, and will serve as a valuable tool in the future development of DENV protease inhibitors.

## Methods

**Expression constructs**. The NS2B construct (comprising residues 47-95, which corresponds to amino acids 1394-1440 in the full-length DENV-2 polyprotein) and the NS3proS135A construct (residues 1-185 corresponding to amino acids 1476-1660 in DENV-2) were generated as described earlier[19,44]. Briefly, the $NS2B_{47-95}$ segment was subcloned into pET21b (Novagen) using the NdeI and BamHI sites. A His6-thrombin protease cleavage site was introduced at the N-terminus of NS2B. A His6 tag was also introduced at the N-terminus of the NS3proS135A domain, and thereafter subcloned into pET21b using the NdeI and XhoI sites. The S135A mutation was introduced using the QuikChange Lightning kit (Agilent, Santa Clara, CA, USA). All cloned constructs and introduced modifications were verified by DNA sequencing. All reagents were from Sigma (St. Louis, MO, USA) unless otherwise stated.

**Protein expression and purification for NMR studies**. The NS2B and NS3proS135A constructs were transformed into *E. coli* T7 express competent cells and expressed separately in different isotopic labelling combinations in[1]/$^2$H, $^{15}$N[12],/$^{13}$C-labelled M9 medium. Chemicals for isotope labelling (ammonium chloride, $^{15}$N (99%), D-glucose, $^{13}$C (99%), deuterium oxide) were purchased from Cambridge Isotope Laboratories Inc. Protein expression was induced for 4-5 h at 37 °C by addition of β-D-1-thiogalactopyranoside (IPTG) to 1 mM final concentration, when the cell optical density at 600 nm ($OD_{600}$) reached 0.8. The cells were thereafter harvested by centrifugation at 6000 *g*.

A methyl protonated Ileδ1-[$^{13}$CH$_3$], Leu, Val-[$^{13}$CH$_3$/$^{12}$CD$_3$], U-[$^{15}$N,$^{13}$C,$^2$H] sample of NS3proS135A was obtained following previous protocols[60]. The protein was expressed in 1 L D$_2$O M9 medium using 3 g/L of U-[$^{13}$C,$^2$H]-glucose as the main carbon

source and 1 g/L $^{15}$NH$_4$Cl (CIL, Andover, MA, USA) as nitrogen source. One hour prior to induction, 70 mg/L of the precursor alpha-ketobutyric acid, sodium salt ($^{13}$C4,98%, 3,3-$^2$H, 98%) and 120 mg/L alpha-ketoisovaleric acid, sodium salt (1,2,3,4-$^{13}$C4,99%, 3, 4, 4, 4,-$^2$H97%) (CIL, Andover, MA, USA) were added to the growth medium. The growth was continued for 2 h at 37 °C and cells were thereafter harvested by centrifugation. NS2B and NS3proS135A were purified separately and refolded as previously described[19]. Briefly, cells were resuspended in 1x PBS buffer and lysed using ultra-sonicator followed by centrifugation at 40,000 g for 30 min. His-NS2B was collected and purified from a Ni$^{2+}$ Sepharose 6 Fast Flow column. The target protein was eluted with 1 x PBS buffer containing 500 mM imidazole. His-NS2B and thrombin protease were added to the dialysis tubing (3.5 kD MWCO) to cleave off the His6 tag from NS2B. The protein was further purified with a size-exclusion chromatography using a HiLoad 16/60 Superdex 200 column, reaching >95% purity as checked by SDS-PAGE. His-NS3proS135A was expressed and purified as inclusion bodies, solubilization was performed overnight on a rolling board in a buffer comprising 8 M urea, 50 mM Tris pH 7.6 (RT), 20 mM imidazole, 0.5 M NaCl. NS2B and His-NS3proS135A were co-refolded by one-step dialysis overnight at 4 °C in a 2:1 molar ratio to maximize the formation of the active complex. The refolding buffer was 25 mM Tris pH 8.5 (pH set at 4 °C), 100 mM NaCl. The NS2B/His-NS3proS135A heterodimer was further purified on a HiLoad 16/60 Superdex 200 size exclusion column (Cytiva). The NS2B/His-NS3proS135A heterodimer was concentrated to 0.4–0.8 mM for data acquisition in NMR buffer containing 20 mM MES pH 6.5, 100 mM NaCl, 5 mM CaCl$_2$, 1x cocktail, 0.02% NaN$_3$ and 10% D$_2$O. Deuterated MES (CIL, Andover, MA, USA) was used for all deuterated NMR samples. The His- part of the NS3proS135A will not be used anymore in the text.

**NMR spectroscopy**. NMR spectra of a methyl protonated Ileδ1-[$^{13}$CH$_3$], Leu, Val-[$^{13}$CH$_3$/$^{12}$CD$_3$], U-[$^{15}$N,$^{13}$C,$^2$H] sample of NS2B/NS3proS135A were recorded at 298 K on 600-900 MHz AVANCE III Bruker spectrometers equipped with 3 and 5 mm cryo-enhanced probes. Data were processed either by NMRPipe[61] or TopSpin 4.06 (Bruker, Billerica, MA, USA), and analysed using CcpNmr2.4.2[62]. and Dynamics Center2.8 (Bruker, Billerica, MA, USA). Backbone and Methyl groups of Val, Leu and Ile resonance assignment for DENV-2 NS2B/NS3proS135A were obtained as previously described[44]. Intra- and inter-molecular distance restraints were extracted from $^1$H–$^1$H Nuclear Overhauser Enhancement (NOE) data collected at 600 MHz with standard Bruker pulse sequence, 3D NOESY-$^{15}$N-HSQC, with a mixing time of 100 ms. Intra- and inter-molecular distance restraints were calibrated using CcpNmr2.4.2[62].. Upper bound distances for NOE restraints were derived from NOE cross-peaks volumes using characteristic distances (sequential NH-NH NOEs in β-sheets or intra-residual NH-Methyl in alanine residues). Methyl-methyl distance restraints, derived from the 4D NUS(15%) $^{13}$C, $^{13}$C-methyl NOESY[63] experiments performed on a spectrometer at 900 MHz were processed by NMRPipe[61] and the IST algorithm in the mddnmr software[64,65] and were all set to 5.5 Å. Backbone dihedral angle restraints were predicted using the TALOS-N software[66] based on chemical shifts of $^1$HN, $^{15}$N, $^{13}$Cα, $^{13}$Cβ, $^{13}$C′ and $^1$Hα nuclei, previously derived for the DENV-2 NS2B/NS3proS135A heterodimer[44] (BioMagResBank accession code 51149). Predictions were converted to dihedral angle restraints with an error corresponding to twice the standard deviations given by TALOS-N. $^{15}$N backbone spin relaxation measurements were performed using sensitivity-

enhanced TROSY-type pulse-sequences, with the addition of temperature compensation train of pulses before acquisition time in $R_1$ and $R_2$ relaxation measurement experiments[67]. 3D pseudo spectra were recorded at a 600 and 700 MHz. For longitudinal relaxation rate ($R_1$) and transverse relaxation rate ($R_2$) measurements of the proton $^1H$, spectral widths SW($^1H$) of 16 ppm over 1024 complex points in the $^1H$ dimension, and 40 ppm over 128 complex points in the nitrogen $^{15}N$, dimension with 24 transients (NS) were used. Relaxation delay (D1) with a duration of 1 s was used before temperature block. The $R_1$ relaxation value was determined from series of 11 relaxation delays including 10, 90, 192, 260, 380, 480, 690, 980, 1220 and 1444 ms with a repetition of delays corresponding to 10 ms. The $R_2$ relaxation value was measured using 12 Carr-Purcell-Meiboom-Gill (CPMG) delays of 8.48, 16.96, 25.44, 33.92, 42.4, 50.88, 59.36, 67.84, 76.32, 84.80 and 110.24 ms with a repetition of delays set at 8.48 ms. Both $R_1$ and $R_2$ experiments were repeated twice to estimate the experimental errors which were below 5%. Backbone ($^1H$)$^{15}N$ steady-state heteronuclear NOEs were measured using TROSY type experiments[67]. 2D experiments consistent from acquisition of NOE-enhanced and an unsaturated spectra were collected using D1 = 3 s, SW($^1H$) = 16 ppm with 1024 complex points in the $^1H$ dimension and SW($^{15}N$) = 40ppm with 256 complex points, NS = 48. NOE values were obtained by dividing $^1H$-$^{15}N$ peak intensities in a NOE-enhanced spectrum by the corresponding intensities in an unsaturated spectrum. An error of about 5% was set for all NOE experiments. All relaxation experiment data were evaluated using Bruker Dynamics Center2.8. The order parameter $S^2$ and the fast internal correlation time $\tau^E$ were obtained by fitting the relaxation parameters $R_1$ and $R_2$, as well as the hetero-nuclear NOE values at two fields, using the Lipari-Szabo model-free approach with a NH bond length of 1.02 Å, and a $^{15}N$ chemical-shift anisotropy value (CSA) of $-166$ ppm[51,52,68].

Relaxation measurements for methyl $^{13}C^1H_3$ groups[50,69] were performed at 800 MHz as a interleave pseudo-3D spectra. Experimental parameters in measurement of longitudinal relaxation rates $R_1$ for $^{13}C^1H_3$ were obtained as previously described[42] by mono-exponential fitting of cross peak intensities in sets of 12 two-dimensional correlation spectra recorded with T1 relaxation delays ranging from 0.01, 0.04, 0.08, 0.13, 0.20, 0.29, 0.41, 0.57, 0.69, 0.99, 1.39 to 2.00 s. The number of transitions was set to 16. Dipolar CH, CH cross-correlation contribution to $R_2$ (named in this study as Γ2) for $^{13}C^1H_3$ groups were measured as previously described[42,69], with a constant time period of 28.6 ms and 14 evolution delays Δ, ranging from 0.01, 0.6, 1.2, 1.8, 2.4, 3.0, 3.6, 4.2, 4.8, 5.6, 6.4, 7.2, 8.0 to 9.2 ms. The number of transitions was set to 32. For both $R_1$ and Γ2 relaxation experiments on $^{13}C^1H_3$ groups, all parameters were set as previously described[42]: the $^1H$ and $^{13}C$ carrier frequencies were set to water resonance, 4.7 and 16.5 ppm, respectively, SW($^1H$), of 12 ppm over 1024 complex points in the $^1H$ dimension and 16 ppm over 80 complex points in the $^{13}C$ dimension, and the inter-scan delays were equal to 1 s. Processing of $R_1$ and Γ2 spectra sets was performed in the TopSpin3.5pl7 software (Bruker) and analysed in the Mathematica software package (Wolfram Research Inc.) as previously described[42].

**Molecular dynamic calculations**. The structure refinement and analysis of NS2B/NS3proS135A were performed using molecular dynamic (MD) simulations, based on the Amber20 software package[70] and parallel computing on central processors (CPU) and graphics accelerators (GPU) with the CUDA hardware and

software architecture, respectively. All MD simulations were performed using the ff14SBonlysc force field with improved side-chain calibration governing intermolecular interactions[71]. The charge of the protein residues was calculated using the pdb2pqr software v 3.1.0[72]. The length of NMR-based restrained MD calculations was 1μs for every conformation. The length of free MD trajectories was chosen to be sufficient for collecting the data for the analyses presented in the next section and was more than 10 times the overall correlation tumbling time of molecule $\tau_c$. Starting structures for free MD and simulation parameters can be found in Supplementary Note S3 and downloaded from the link in the data availability section. MD simulation trajectories were analysed by Hierarchical cluster analysis using the cpptraj software of Amber20[73].

**Individual MD trajectory analyses with correlation function down-sampling**. MD trajectory analyses with back-calculation of NMR spin-relaxation parameters for NH vectors were utilized as previously described[40], with the aid of the "Mathematica" software package [Wolfram Research] and the MD Analysis external library [mdanalysis.org]. The starting point of our analyses was the alignment of all protein frames onto the mean structure (for each corresponding MD trajectory), using heavy atoms from rigid backbone regions corresponding to the stretch of residues 47–87 in NS2B and 10–170 in NS3proS135A. This alignment step removes translational and rotational diffusion from MD trajectories. The auto-correlation functions C(t) for the XH bond (where X can be $^{15}N$, $^{13}C$ or $^1H$) were calculated using normalized XH($\tau$) and XH($\tau + t$) vectors according to equation 2 $C(t) = \langle P_2[XH(\tau)XH(\tau + t)]\rangle$ in[40] whereas for XH, XH' cross-correlation functions of two vectors XH($\tau$) and XH'($\tau + t$) were used. Moreover, the cross-correlation functions for the methyl groups were averaged over all CH vector pairs of the corresponding methyl group. Similarly, the cross-correlation functions were averaged over all HH' vector pairs. The length of correlation function C(t) was calculated for the time intervals up to $7\tau_c$ was set to 145 ns (where $\tau_c$ is the overall rotational correlation of the molecule in solution) that was according to ref. [40]. For the subsequent multi-exponential approximation of the correlation functions C($t_i$), we used a weighted function W($t_i$) proportional to $|C(t_i)-C(t_{i+1})|/\sigma_i$, where $\sigma_i$ is a bootstrapping standard deviation of C($t_i$) described below. This allowed both filtration and down-sampling of C(t). More specifically, to obtain N points of the correlation function after down-sampling, the starting down-sampling point $t_{dsstart}$ was derived from the following equation:

$$|C(t_{dsstart}) - C(t_{dsstart-1})| = |C(t_{dsstart}) - C(t_{final})|/(N - dsstart)$$
$$= |C(t_k) - C(t_{k+1})|$$

for all k from $d_{sstart}$ to N.

The values of the correlation function C($t_k$) in the center of the down-sampled $k^{th}$ interval were calculated from the classical second-degree polynomial filter[74–76]. We tested the number of C($t_k$) points N in ranges from to 2 K till 32 K, which resulted in only insignificant differences compared to the back-calculated NMR relaxation parameters. Therefore, the N value was set to 2 K. The down-sampling for cross-correlation was performed in a similar way. Examples of auto- and cross-correlation function down-sampling are presented in Fig. 4.

**Calculation of theoretical NMR spectral densities from correlation functions**. The down-sampled auto-correlation function was fitted to a multi-exponential decay $C(t) = A_0 + \sum_{j=1}^{m} A_j e^{-t/\tau_j}$

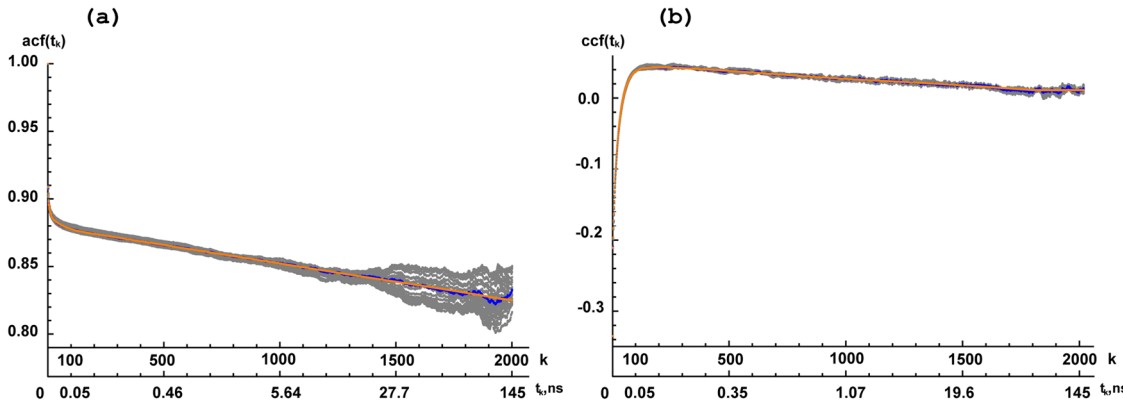

**Fig. 4 Examples of auto- and cross-correlation functions down-sampling and approximation.** Panel (**a**) displays the auto-correlation function, acf(t$_k$), of 75 S NS2B NH vectors for trajectory I. Panel (**b**) presents the cross-correlation CHCH' function, ccf(t$_k$), for 73ICD1 methyl group vectors for NS2B trajectory I. The abscissa axes represent the correlation function values, whereas the two ordinates axes represent the point number k after down-sampling and the time t as a function of k. The blue curves present data obtained from the calculated MD trajectories. The average of the results of four exponential approximations are represented by the yellow curves. The bootstrapping statistical analysis (subset of 32 curves, each using one-half fraction of MD trajectory) are shown in grey.

as described in equation 9 in[40]. The exponent number *m* for each correlation function was tested from 2 to 7. An approximation was performed in the Mathematica program utilizing the Levenberg-Marquardt minimization approach as described in the Supplementary Note S4 section. Representative fits of different auto- and cross-correlations to multi-exponential models are presented in Fig. 4. Finally, the best-fit parameters A$_0$, A$_j$, and τ$_j$ provide spectral density functions according to equation 10 in[40]:

$$J(\omega) = \frac{A_0 2\tau_c}{1 + (\omega\tau_c)^2} + \sum_{j=1}^{m} \frac{A_j 2\tau_j'}{1 + (\omega\tau_j')^2},$$

where $\tau_j' = \tau_c\tau_j/(\tau_c + \tau_j)$ $\tau_c$ is the experimental rotation correlation time.

**Back-calculation of theoretical relaxation parameters from spectral density functions.** For $^{15}$N uniformly-labelled proteins, classical NMR $^{15}$N relaxation rates R$_1$, R$_2$ and heteronuclear NOE were provided as a function of spectral densities[40,77,78]. For the methyl $^{13}$C$^1$H$_3$ groups in the ($^{15}$N/$^{13}$C/$^2$H)-$^{13}$CH$_3$-labelled protein, the longitudinal relaxation rate R$_1$ and the dipole-dipole cross-correlation contributions to R$_2$ (Γ$_2$) were calculated using the following equations previously described in[50,69,79]:

$$R_1 = \frac{3}{20}\left(\frac{\mu_0 h \gamma_H \gamma_C}{8\pi^2 r_{CH}^3}\right)^2 \left[J(\omega_H - \omega_C) + 3J(\omega_C) + 6J(\omega_H + \omega_C)\right]$$
$$+ \frac{1}{20}\left(\frac{\mu_0 h \gamma_C \gamma_C}{8\pi^2 r_{CC}^3}\right)^2 \left[J(0) + 3J(\omega_C) + +6J(2\omega_C)\right] + \frac{\omega_C^2 \Delta\sigma^2}{15} J(\omega_C)$$

$$\Gamma_2 = \frac{1}{20}\left(\frac{\mu_0 h \gamma_H \gamma_C}{8\pi^2 r_{CH}^3}\right)^2 \left[4J_{CH,CH'}(0) + 3J_{CH,CH'}(\omega_C)\right]$$
$$+ \frac{1}{20}\left(\frac{\mu_0 h \gamma_H^2}{8\pi^2 r_{HH}^3}\right)^2 \left[3J_{HH',HH''}(\omega_H) + 3J(2\omega_H)\right],$$

where $\mu_0$ is the vacuum permeability; *h* is Planck's constant; $\gamma_H$ and $\gamma_C$ are the gyromagnetic ratios of $^1$H and $^{13}$C, respectively; $\Delta\sigma$ is the chemical shift anisotropy (CSA) of $^{13}$C = 25ppm and $^{15}$N CSA = −166ppm[80]; $r_{NH}$ = 1.016Å[81], for CH, CC' and HH distances in methyl group are $r_{CH}$ = 1.07Å, $r_{CC}$ = 1.533Å; and

$r_{HH}$ = 1.80Å[82], respectively; $\omega_H$ and $\omega_C$ are Larmor frequencies of spins $^1$H and $^{13}$C at 800 MHz, respectively; *J* is the auto-correlation spectral density; ' and $J_{HH',HH''}$ are cross-correlation spectral densities for pairs CH-CH' and HH'-HH'' according to[50,69], respectively. To exclude the R$_1$ systematic errors arising as a result of the methyl rotation torsion potential uncertainties discussed in detail in[83] the methyl MD back-calculated R$_1$ values were normalized *versus* the experimental values.

**Statistics and reproducibility**
*Block bootstrapping statistical analysis of back-calculated relaxation parameters.* The traditional analysis of a set of MD trajectories under the same conditions, known as classical bootstrapping, involves using random subsets of the MD trajectories. However, in our study, the primary focus of the manuscript is the development of a method for ranking individual MD trajectories that correspond to unique starting structures. These starting structures can be obtained through various methods including MD annealing results, NMR-derived structures, single X-ray structures, Cryo EM structures and even from AlphaFold prediction structures. Consequently, the dispersion of the back-calculated parameters need to be estimated within each individual trajectory corresponding to the single structure. This is in contrast to the tradition method of error estimation. To address this, we employed a moving block bootstrapping procedure with overlapping blocks[84,85] to estimate the dispersion of the back-calculation parameters from individual MD trajectories. As mentioned above the equation C(t)=⟨P₂[XH(τ)XH(τ + t)]⟩ was used for correlation function back-calculation with averaging over all τ values, whereas a bootstrapping used averaging over a half of τ values. The 32 bootstrapping blocks were used ranging from (0; τ$_{maximum}$/2) till (τ$_{maximum}$/2; τ$_{maximum}$) resulting in 32 correlation functions C(t) presented in grey in Fig. 4. All 32 correlation functions undergo the same procedures as the main one: down-sampling preserving same t$_k$ points, multi-exponential approximation, and the subsequent 32 relaxation parameter back-calculated. The standard deviation of the 32 relaxation parameter values divided by √2 were used as a discrepancy of the value in the particular single trajectory. The final errors (presented in figures N, N + 1, N + 2…) of back-calculated relaxation parameters were a geometric mean of bootstrapping discrepancy described above and

systematic errors estimated from the rNH, rCH and $^{15}$N or $^{13}$C CSA uncertainties.

**Reporting summary**. Further information on research design is available in the Nature Portfolio Reporting Summary linked to this article.

## Data availability

The starting structures that supports the findings in this study and the numerical sources for the graphs are available from: https://doi.org/10.5878/5wea-fk76. Other information and data are available upon request from the authors.

## Code availability

For NMR TopSpin3.5pl7, TopSpin4.0.6 (Bruker.com) and NMRPipe (NMRbox.org) were used. For MD calculations Amber20 (ambermd.org). For analysis of MD: Mathematica (wolfram.com) with the MD Analysis external library (mdanalysis.org). Other information is available upon request from the authors.

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

## Acknowledgements

The authors thank the Swedish NMR centre for access to the instruments and support. This work was supported by the Swedish Foundation for Strategic Research grant ITM17-0218 to T.A and P.A., grant RSF 23-44-10021 to D.M.L., project 121112900217-3 to A.A.L., Swedish Cancer Society grant 21 1605 Pj01H to A.A., and the Swedish Research Council grants 2021-05061 to A.A. and 2019-03561 to V.O. This study used NMRbox: National Center for Biomolecular NMR Data Processing and Analysis, a Biomedical Technology Research Resource (BTRR), which is supported by NIH grant P41GM111135 (NIGMS).

## Author contributions

X.H. and R.S. have contributed to the production of all necessary protein variants, their purification, and developing a construct of labelled DENV II proteins. T.A. wrote the original manuscript draft. P.A. and A.A. contributed to the writing, reviewing, and final editing of the manuscript. T.A. and P.A. performed the NMR studies on DENV II. V.O., D.L. contributed with NMR measurements and methodology, developed and performed statistics analysis and back-calculation of relaxation parameters from MD trajectories making a concept of conformational filter. T.A. performed assignments using the ccpn program. A.L. performed MD calculations. T.S. did the analysis of earlier crystallographic data. P.A., T.A., T.S., A.A., and V.O. conceptualized the project, supervised different parts of the project and acquired the necessary funding acquisition.

## Funding

## Competing interests

The authors declare no competing interests.

**Additional information**

