## [Peer Review File · Communications Biology]

Reviewers' comments:

Reviewer #1 (Remarks to the Author):

Agback et al. reported a study to evaluate the conformation/structure of dengue protease. due to the importance of this protease in maturation of viral proteins, the protease is a validated target for drug discovery. The structural information in this study is useful for structure-guided drug discovery. the closed conformation of dengue protease is the main the confirmation while open form was observed in some crystal studies due to several factors such as crystal packing or the presence of the glycine linker in the construct. .

overall this manuscript is well written and the follow minor sections need to be corrected or considered.

1> please include one to two sentences to explain the how the protease was purified (e.g refolding or purifying using a solution form).

2> Please confirm the delays using in R2 data collection, is ms or s?

3> in MD simulation, does the NOEs used include restraints favoring the closed conformation? can the confirmation be confirmed based on the NOEs?

4> relaxation data are quite useful. it will be good to include a data set with an inhibitor/peptide substrate. It may not be possible to perform such an experiment, a comparison or discussion with a published data set in which an inhibitor is included will be useful.

5> Please strength the advantages of using back-calculated dynamics parameters and the reason to use this strategy to emphasize the novelty of this manuscript.

Reviewer #2 (Remarks to the Author):

The paper brings a very interesting approach to overcome the limitations of crystal structures using a combination of NMR+MD simulations. I really believe this novel type of combinations can interrogate dynamic proteins in a very interesting manner, however the manuscript could use a longer discussion and contextualization on previous work done in the target (DENV protease) as well as some quality of life illustrations/explanations for a more general reader. Specially where this approach could be applied for other proteins.

Specifically, some minor details could also be addressed:

1- Please comment on the timescale of the cited simulations (Refs 26-28) and which conclusions could be drawn from those papers. Also there is newer literature with longer simulations which should be cited and discussed.

"This hypothesis would also imply the simultaneous presence of open and close conformations, with a possible exchange time scale of 10 ms"

2- from where did the authors conclude that this is the timescale of the changes? could they elaborate on that?

(it would be interesting to discuss the changes and timescale of changes against NMR studies from the literature).

3- Could the authors comment on why the Paragraph below is highlighted?

"Since the three-dimensional structure of the apo form of DENV-2 NS3pro/NS2B complex in solution remained missing, we

performed a detailed NMR investigation starting with the apo form of the Ser135Ala (NS3proS135A) mutated protein variant. This mutation abolishes protease activity with minimal interference on the overall three-dimensional structure⁴²...."

"Our results demonstrate that the choice of adequate ensembles of potential conformations for large proteins, combined with the use of specific force fields and stringent NMR data allows us to unambiguously probe the existence of different conformations for NS2B/NS3."

4- what exactly do the authors mean by specific force fields, were there more than one tested in this study?

How do their protocol compares against literature data on the NS2B/NS3 simulations?

"We believe that our results, based only on unlinked full-length NS2B/NS3pro protease heterodimers demonstrate that the so-called 'open' inactive form of NS2B/NS3pro is mainly due to crystallographic packing."

5- could the authors elaborate, perhaps showing those crystal packing artifacts as a SI-example?, on these problems

as a suggestion, some sort of cartoon or scheme discussing the sampling expansion, which conditions and how many sub replicas would help a naive reader.

(how many 1 μ s simulations and how many 250ns out of those).

Should the authors use PCA for clusterizing the trajectories would the 4 clusters be similar (3+open) or would they rather be distinct, which movements and pockets are unique for each of the three identified conformations? how do they converse with previously identified unique conformations?

Referee expertise:

Referee #1: NMR

Referee #2: MD simulations

Reviewers' comments:

Reviewer #1 (Remarks to the Author):

Agback et al. reported a study to evaluate the conformation/structure of dengue protease. due to the importance of this protease in maturation of viral proteins, the protease is a validated target for drug discovery. The structural information in this study is useful for structure-guided drug discovery. the closed conformation of dengue protease is the main the confirmation while open form was observed in some crystal studies due to several factors such as crystal packing or the presence of the glycine linker in the construct. . overall this manuscript is well written and the follow minor sections need to be corrected or considered.

1> please include one to two sentences to explain the how the protease was purified (e.g refolding or purifying using a solution form).

The purification description have been added to materials and methods. We somehow managed to accidentally remove it when writing the manuscript, sorry about that.

2> Please confirm the delays using in R2 data collection, is ms or s?

Text changed so that ms is showed for all delays that was used.

3> in MD simulation, does the NOEs used include restraints favouring the closed conformation? can the confirmation be confirmed based on the NOEs?

In this publication we aimed to explore whether the measured relaxation parameters could effectively differentiate between MD-trajectories. We assert that this is indeed the case. Therefore, the primary objective of this manuscript is to rank MD-trajectories based on their alignment with experimental relaxation data. The proposed method in this study should be considered as a complementary alternative to currently existing or under development methods, including the widely used NOE based methodology. Thus far, no methodologies are without problems. Such discussions are better suited for a review publication rather than a research manuscript. Nevertheless, we can present a few statements that inspire us to search to other alternative approaches in addition to the NOE based methodology.

a. Typically, the intermolecular 1H-1H NOE cross-peaks exhibit a lower signal-to-noise ratio compared to the intramolecular NOE cross-peaks and, importantly, even less than the diagonal ones. In contrast, the signal-to-noise ratio in the relaxation experiments is of the same order as the diagonal peaks. These relaxation experiments are designed to overcome the NMR limitations imposed by the system, unlike the intermolecular 1H-1H NOE cross-peaks.

b. Moreover the 1H-1H NOE cross-peaks are seldom reliable when back-calculated from MD-trajectory due to the cooperativity (in contrast to 15N1H and 13C1H3 relaxation data) of the 1H-1H cross-relaxation NOE peaks which involve spin diffusion, spectral densities as a result of 1H-1H distance variation, chemical exchange (Rex) in both direct and indirect dimensions in nD NOE spectra, exchange amide protons with water.

c. So, the observed intermolecular NOEs do indeed support the closed conformation. However, it is important to note that NOEs can only provide a positive answer, not a negative one.

4> relaxation data are quite useful. it will be good to include a data set with an inhibitor/peptide substrate. It may not be possible to perform such an experiment, a comparison or discussion with a published data set in which an inhibitor is included will be useful.

This will be our next step. Experiments are ongoing. There we will have more structural data as the complex is less dynamic than the apo form. In this study, our focus is on the highly dynamic apo form. It should be noted that the ligands we are using, which are commonly used as serine protease inhibitors, contain boronic acid or CF3 groups that require parametrization in the force field. However, this was obviously not necessary for the apo form in this stage of our research.

5> Please strength the advantages of using back-calculated dynamics parameters and the reason to use this strategy to emphasize the novelty of this manuscript.

As mentioned earlier (in response to question 3), there are numerous challenges associated with NOEs. Therefore, in many cases, it proves beneficial to utilize NMR relaxation data for ranking conformations obtained in silico. In a previous study (Lesovoy, D. M. et al. NMR relaxation parameters of methyl groups as a tool to map the interfaces of helix-helix interactions in membrane proteins. Journal of Biomolecular NMR 69, 165-179, doi:10.1007/s10858-017-0146-1 (2017)), one of the authors participating in this study employed experimental methyl group relaxation data to rank docking results for dimerization. In our current work, we utilize experimental backbone and side-chain relaxation data to rank all-atom MD trajectories. Furthermore, we elucidate which backbone and side-chain relaxation parameters exhibit the highest sensitivity and suitability for back-calculation, enabling subsequent comparison with experimental data to establish criteria for ranking MD trajectories. To underscore the reliability of the relaxation parameters back-calculated from MD

simulations, we have introduced a dedicated subsection titled "Block bootstrapping statistical analysis of back-calculated relaxation parameters"

The text have been changed in several places to show the advantages of our proposed method. We have also added a graphic representation of the method.

Reviewer #2 (Remarks to the Author):

The paper brings a very interesting approach to overcome the limitations of crystal structures using a combination of NMR+MD simulations. I really believe this novel type of combinations can interrogate dynamic proteins in a very interesting manner, however the manuscript could use a longer discussion and contextualization on previous work done in the target (DENV protease) as well as some quality of life illustrations/explanations for a more general reader. Specially where this approach could be applied for other proteins. -

We did a rewrite of the introduction and hope it is clearer now. See also answer >5 for Reviewer 1

Specifically, some minor details could also be addressed:

>1- Please comment on the timescale of the cited simulations (Refs 26-28) and which conclusions could be drawn from those papers. Also there is newer literature with longer simulations which should be cited and discussed.

This references (Refs 26-28 numbering in submitted version) are related to the MD simulation complex of the protein with ligand. Those references were mentioned in introduction of this study solely to illustrate that inhibitor can induce a different conformations. In this paper we are exclusively focused on the apo form. In our upcoming paper, where we will be utilizing inhibitor, we will engage in more in –depth discussion.

Additionally, we have included relevant recent references that we have found:

- a. Gangopadhyay, A. & Saha, A. Exploring allosteric hits of the NS2B-NS3 protease of DENV2 by structure-guided screening. *Comput Biol Chem* 104, 107876, doi:10.1016/j.compbiolchem.2023.107876 (2023)
- b. da Costa, R. A. et al. A Computational Approach Applied to the Study of Potential Allosteric Inhibitors Protease NS2B/NS3 from Dengue Virus. *Molecules* 27, doi:ARTN 4118
- c. Purohit, P., Barik, D., Agasti, S., Panda, M. & Meher, B. R. Evaluation of the inhibitory potency of anti-dengue phytochemicals against DENV-2 NS2B-NS3 protease: virtual screening, ADMET profiling and molecular dynamics simulation investigations. *Journal of Biomolecular Structure & Dynamics*, doi:10.1080/07391102.2023.2212798 (2023).

In our study, the minimum trajectory duration is approximately $7 \cdot T_c$, i.e. 250 ns. Our intention was to demonstrate that only a minimum trajectory was necessary for testing the dynamics within this local minimum. The objective was to avoid spending too much computation while still being able to discard conformations that do not compatible with the measured relaxation parameters.

That being said, we have now revised the method description and the resulting data to reflect a longer trajectory. This allows us to perform a more comprehensive statistical error analysis. Importantly, this adjustment did not alter the results of the ranking trajectories.

>2- "This hypothesis would also imply the simultaneous presence of open and close conformations, with a possible exchange time scale of 10 ms" from where did the authors conclude that this is the timescale of the changes? could they elaborate on that?

The sentence mentioned above has been slightly modified. Instead of specifying "10ms", it would be more appropriate to refer to "slow exchange in the millisecond range". We have also included a reference to a recent paper that extensively investigated this exchange properties of NS2B in Dengue 2 apo form. This is why we have removed through text all mentions about slow Rex obtained by us in this study

Lee, W. H. K., Liu, W., Fan, J. S. & Yang, D. W. Dengue virus protease activity modulated by dynamics of protease cofactor. *Biophysical Journal* **120**, 2444-2453, doi:10.1016/j.bpj.2021.04.015 (2021)

> 2 (it would be interesting to discuss the changes and timescale of changes against NMR studies from the literature).

We believe that we wrote some discussion: see in Discussion section:

"Indeed, the largest differences are observed for the backbone 1H-15N hetero NOE of Glu80E and Ser83S in NS2B stretching ca 0 versus 0.6 for the 'fully open' conformation versus the experimental values. This result allows us to conclude that if this conformation cannot be fully ruled out as irrelevant only due to the low score of fit, still this possible population can be estimated to be below the level of experimental error, ca 5%. Unfortunately, the present state of MD computation does not allow us to observe the slow transition between higher populated states and the lower populated ones, and this should be considered as a challenge for future development."

And in the next section of the text as well.

3- Could the authors comment on why the Paragraph below is highlighted?

"Since the three-dimensional structure of the apo form of DENV-2 NS3pro/NS2B complex in solution remained missing, we performed a detailed NMR investigation starting with the apo form of the Ser135Ala (NS3proS135A) mutated protein variant. This mutation abolishes protease activity with minimal interference on the overall three-dimensional structure42...."

Sorry about that. The italics was used when we wrote the manuscript and should have been removed. It fills no function in the text and have been removed.

>4- "Our results demonstrate that the choice of adequate ensembles of potential conformations for large proteins, combined with the use of **specific force fields** and stringent NMR data allows us to unambiguously probe the existence of different conformations for NS2B/NS3."

what exactly do the authors mean by specific force fields, were there more than one tested in this study?

The sentence was badly written, sorry for that. We did not mean to indicate that we did a test of force fields. We meant that we chose a force field that we thought was suitable for the task. The text has been changed to: "Our results demonstrate that the choice of adequate ensembles of potential conformations for large proteins, combined with the use of a force field suitable for the task and stringent NMR data allows us to unambiguously probe the existence of different conformations for NS2B/NS3."

Nevertheless, in our text we discuss how we estimate that the choice of force field used in the study is suitable. See Page 27 "Nevertheless, the validity of our back calculation protocol used in this study was evaluated by comparing the theoretically obtained values against the corresponding experimental relaxation parameters located in the first β -barrel of NS3proS135A, between residues 20-28 and 35-60 (**Figures 3 A-C**)."

A recent publication, which was published after our submission, supports our approach. Reference added:

Kummerer, F., Orioli, S. & Lindorff-Larsen, K. Fitting Force Field Parameters to NMR Relaxation Data. *Journal of Chemical Theory and Computation*, doi:10.1021/acs.jctc.3c00174 (2023).

However saying that, testing of different force fields is in our plans.

>How do their protocol compares against literature data on the NS2B/NS3 simulations?

We have included recent papers related to free MD simulation of NS2B/NS3 complexes (as seen above in question 1 reviewer 2). However, we have not delved into the differences in MD protocols in this study. Our primary objective in this research is to compare the calculated result with our experimental data. Nevertheless, it is part of our future plans to explore variations in force fields and protocols, where we will also discuss the methodological differences.

>5- "We believe that our results, based only on unlinked full-length NS2B/NS3pro protease

heterodimers demonstrate that the so-called 'open' inactive form of NS2B/NS3pro is mainly due to crystallographic packing." could the authors elaborate, perhaps showing those crystal packing artefacts as a SI-example?, on these problems .

We have added a new figure to the supplementary information which address this issue. We show that the open conformation can be due to contacts with symmetry related molecules in the crystal. We also discuss this issue in the introduction.

>as a suggestion, some sort of cartoon or scheme discussing the sampling expansion, which conditions and how many sub replicas would help a naive reader. (how many 1 μ s simulations and how many 250ns out of those).

Thanks for the suggestion we have constructed a cartoon/scheme explaining our method. In addition, we inserted a methods subsection titled "Block bootstrapping statistical analysis of back-calculated relaxation parameters" paying a special attention to 32 block bootstrapping sub replicas of NMR relaxation parameters back-calculation from MD-trajectory, producing the errors(added to all the figures and figure legends) of relaxation parameter values.

>Should the authors use PCA for clusterizing the trajectories would the 4 clusters be similar (3+open) or would they rather be distinct, which movements and pockets are unique for each of the three identified conformations? how do they converse with previously identified unique conformations?

PCA, similar to diffusion maps, is a method for reducing the dimensionality of MD trajectories. While these techniques undoubtedly allow to compare dominant modes of motion among different trajectories, they do not directly facilitate comparisons with experimental parameters of NMR relaxation.

However, we performed PCA, and resulting clusters are distinct. We are cautious about the level of detail one can be extracted from our calculations and resented in this particular manuscript. The NS3/NS2Bcomplex undergoes substantial dynamic, and we cannot be certain that we have sampled all or enough of the possible conformations. So far we just rejected the one which are not fitted to the experimental data, so to speak: narrow down the possible conformation space, to demonstrate what filter is doing.

When we apply our method to the NS3/NS2B complex with ligands (work in progress) PCA will likely be used to highlight the anticipated changes upon binding.

However, should the referees express an interest in viewing PCA movies, we are prepared to provide them.

Once again, our primary goal is to demonstrate that the NMR-MD filter proposed in this publication can effectively narrow down and rank the conformation space of protein. The analysis of structural and conformational dynamics is an issue and will be addressed in a future publication.

If, in the future, either our group or others propose new conformations, we plan to perform a new set of calculations to assess their compatibility with our measured relaxation parameters. Those that are not compatible should be discarded, while those that align with experiment should be included as potential conformations of apo NS3/NS2B.

REVIEWERS' COMMENTS:

Reviewer #1 (Remarks to the Author):

The manuscript has been revised by the authors. Suggest for publication.

Authors may consider the following minor items in the proof stage.

- > title: "NS3pro/NS2B dengue-associated protease" is misleading. it is up to the authors to consider using more common description such as dengue NS2B/NS3pro protease.
- > Authors should use more commonly used definition of the protease to make audience to search the paper easily. NS3pro/NS2B might be used as NS2B/NS3pro or NS2B-NS3pro. it is up to author to define the format.
- >R1 and R2, please make sure numbers are shown as Subscript format.

Reviewer #2 (Remarks to the Author):

most of my concerns were discussed by the authors.

I am looking forward reading the next paper where they will also include the inhibitor in their studies.

We appreciate the time and effort taken by the reviewers to review our manuscript. It was valuable comments and questions throughout the process and we feel that they have helped to improve the manuscript.

REVIEWERS' COMMENTS:

Reviewer #1 (Remarks to the Author):

The manuscript has been revised by the authors. Suggest for publication.

Authors may consider the following minor items in the proof stage.

> title: "NS3pro/NS2B dengue-associated protease" is misleading. it is up to the authors to consider using more common description such as dengue NS2B/NS3pro protease.

We have changed it as suggested.

> Authors should use more commonly used definition of the protease to make audience to search the paper easily. NS3pro/NS2B might be used as NS2B/NS3pro or NS2B-NS3pro. it is up to author to define the format.

We have now changed to NS2B/NS3pro throughout the manuscript.

>R1 and R2, please make sure numbers are shown as Subscript format.

Changed as suggested.

Reviewer #2 (Remarks to the Author):

most of my concerns were discussed by the authors.

I am looking forward reading the next paper where they will also include the inhibitor in their studies.

We are working on the study with inhibitors and are planning to write it up during the first half of 2024. We hope you will find it interesting to read when published.